# Lignin Microspheres Modified with Magnetite Nanoparticles as a Selenate Highly Porous Adsorbent

**DOI:** 10.3390/ijms232213872

**Published:** 2022-11-10

**Authors:** Vesna Marjanovic, Radmila Markovic, Mirjana Steharnik, Silvana Dimitrijevic, Aleksandar D. Marinkovic, Aleksandra Peric-Grujic, Maja Đolic

**Affiliations:** 1Mining and Metallurgy Institute Bor, Zeleni Bulevar 35, 19210 Bor, Serbia; 2Faculty of Technology and Metallurgy, University of Belgrade, Karnegijeva 4, 11000 Belgrade, Serbia

**Keywords:** natural polymer, iron oxide (Fe_3_O_4_), selenium, adsorption, green chemistry

## Abstract

Highly porous lignin-based microspheres, modified by magnetite nanoparticles, were used for the first time for the removal of selenate anions, Se(VI), from spiked and real water samples. The influence of experimental conditions: selenate concentration, adsorbent dosage and contact time on the adsorption capacity was investigated in a batch experimental mode. The FTIR, XRD, SEM techniques were used to analyze the structural and morphological properties of the native and exhausted adsorbent. The maximum adsorption capacity was found to be 69.9 mg/g for Se(VI) anions at pH 6.46 from the simulated water samples. The modified natural polymer was efficient in Se(VI) removal from the real (potable) water samples, originated from six cities in the Republic of Serbia, with an overage efficacy of 20%. The regeneration capacity of 61% in one cycle of desorption (0.5 M NaOH as desorption solution) of bio-based adsorbent was gained in this investigation. The examined material demonstrated a significant affinity for Se(VI) oxyanion, but a low potential for multi-cycle material application; consequently, the loaded sorbent could be proposed to be used as a Se fertilizer.

## 1. Introduction

The industry is generating a large amount of diverse types of wastewater containing harmful and toxic chemicals [1]. Selenium is a trace element in natural deposits of ore containing sulfides of heavy metals, but there are no geologic formations in which selenium is a major constituent. The major sources of selenium are black shale, phosphate rocks, coal, and limestone [2]. The contamination of the environment by selenium compounds can originate from various sources including agricultural drainage water, mine drainage, residues from fossil fuel, thermoelectric power plants, oil refineries, and metal ore.

Selenium is an analog to sulfur and exists naturally in four oxidation states including selenide (-II), selenium (0), selenite (IV), and selenate (VI). The two most common inorganic forms of Se in natural water are selenite (SeO_3_^2−^) and selenate (SeO_4_^2−^). Selenite, or Se(IV), is the most toxic of the four states due to its higher bioavailability resulting in its toxicity being 10 times higher than Se(VI). Generally, Se(VI) dominates anthropogenic discharges into aquatic ecosystems, while Se(IV) is the typically dominant species in discharges such as the fly ash of coal-fired power plants and oil refining wastewater [3]. Both selenite and selenate can appear in a protonated or deprotonated form, depending on the solution pH [4].

According to the World Health Organization (WHO), the maximum daily intake of selenium should not exceed 70 μg/day [5] Selenium deficiency in the diet may have an adverse effect on health. However, its toxic properties should not be forgotten—especially considering its narrow therapeutic index, as its toxic dose starts at 400 μg/day [6]. The toxic effect of selenium depends on its chemical form and concentration in the aqueous systems. The provisional WHO guidelines for selenium in drinking water was set at 40 μg/L [7], while the current common limit of 10 μg/L is applied for selenium in drinking water in most countries or regions [8]. 

Although there are several technologies available for removing Se from water, the use of adsorbents based on the natural materials, such as polymers from biomass, hemicellulose, and lignin for the wastewater treatment is increasing due to their economy, large quantities in paper production, and ability to remove the pollutants. Lignin is generally the most complex and smallest fraction, representing about 10–25% of the biomass by weight. Lignin is a complex, three-dimensional, amorphous, cross-linked phenolic based polymer [9]. A large number of the -OH groups make lignin suitable for functionalization by various chemical methods, in order to adapt its structure, morphological and adsorption characteristics [10], and lignin can be a good starting point to predict the efficient industrial application of a new green for its treatment and valorization [11].

The bio-sorbents for selenium removal from water samples previously tested are peanut shell [12], rice husk [13], green macroalgae [14,15], dried biomass of baker’s yeast *Saccharomyces cerevisiae* [16], fish scale [17], *Ganoderma lucidum* [18], fungal—*Ascomycota* [19], and aquatic weeds [20], among others. Plant (lignocellulosic) materials are the most promising raw material as natural and renewable resources; a significant amount of these materials are the waste by-products generated in various parts of the agricultural industry [21].

Magnetite (Fe_3_O_4_) is one of the mostly widespread iron (hydro)oxide minerals in nature, and is also known as a sink for dissolved oxyanions due to its high adsorption and reduction potential. Due to these properties and their magnetic character, Fe(III) oxides and hydroxides are important sorbents for highly mobile (SeO_4_^2−^) in soils and sediments [4]. The sorption of selenite (SeO_3_^2−^) and selenate (SeO_4_^2−^) from aquatic media onto Fe_3_O_4_ nanomaterials, produced by the (non) microwave-assisted synthetic techniques, was investigated through a batch technique [22]. Compared with either natural magnetite (<5 μm) or nano-iron (~10 nm), the nano-magnetite particles (10–20 nm) were found to be a better adsorbent for selenite, while the nano-iron had better adsorption performance for selenate [23]. 

In this work, the amino-functionalized lignin microspheres (A-LMS) modified with magnetite nanoparticles (NP-Fe_3_O_4_), named A-LMS_Fe_3_O_4_, which have already been used for arsenate and chromate separation [24], were applied to the separation of SeO_4_^2−^ from synthetic and real water samples. The aim was to study this natural, readily available material as a low-cost alternative to traditional adsorbents.

## 2. Results and Discussion

### 2.1. Material Synthesis and Design 

The synthesized material for the Se(VI) anion removal was named as follows—amino-functionalized lignin microspheres modified with magnetite nanoparticles: A-LMS Fe_3_O_4_ [24].

### 2.2. The Effect of Adsorbent Dose 

The effect of adsorbent dose on the adsorption efficiency and adsorption capacity was investigated with an initial concentration of Se in the solution of 7.75 mg L^−1^, pH 6.46, temperature 22 °C and a stirring speed of 170 rpm^−1^, solution volume 11 mL. This is in line with the available data in literature for the Se concentration in wastewater from the flue gas desulfurization process (1–10 mg L^−1^), as well as lead smelters (3–7 mg L^−1^) [2,3,25].

The adsorbent dose was 0.11, 0.23, 0.46, 0.67, and 0.92 g L^−1^. The dependence of adsorption efficiency and adsorption capacity on adsorbent dose is shown in Figure 1. It was shown that with an increase in mass of adsorbent, there was no significant change of adsorption efficiency, indicating the favorable usage of the lower amount of material in a real application. The adsorbent dose of 0.23 g L^−1^ was adopted as optimal due to the efficiency of Se removal greater than 99.8%. High adsorption efficiency (>99%) of the A-LMS Fe_3_O_4_ adsorbent can be explained by a high porosity and many free active sites.

### 2.3. The Effect of Selenium Concentration

The effect of the initial concentration of selenium in simulated water samples on adsorption efficiency and adsorption capacity was done with the following experimental conditions: an initial concentration of Se in the solution of 3.91, 7.75, and 9.27 mg L^−1^, temperature 22 °C and mixing speed of 170 o min^−1^, adsorbent dose of 0.23 g L^−1^, the obtained data are shown in Figure 2. The optimal removal rate was achieved at 99.47% under an initial concentration of 7.75 mg L^−1^, which is further used in the kinetic studies.

The effect of initial adsorbate concentration on the amount of adsorbed Se(VI), was studied by varying the initial concentration of Se in the range of 3–10 mg L^−1^. The increase in initial Se(VI) concentration, increased the amount of adsorbate onto the surface (till the equilibrium concentration was achieved; the sorption capacity was met at 69.9 mg g^−1^), Figure 3. 

### 2.4. Kinetic Studies

The correlation of sorption capacity versus contact time is shown in Figure 4. After the process initiation and within the first 50 min, the removal was rapid, which may be related to the higher number of available activated sites, as well as to the more intensive driving force for the mass transfer. The obtained sorption capacity in equilibrium was around 33.94 mg g^−1^, while the equilibration was attained after 300 min.

Higher *R*^2^ values of described models (Table 1) may point out the most suitable models for data prediction. The plot *t*/(*q_t_*) vs. *t* is shown in Figure 5. The straight line indicates that selenate adsorption by the A-LMS Fe_3_O_4_ adsorbent may be described by the pseudo-second order kinetic model. Comparing the experimentally determined *q*_e_ values and those obtained from the plot of *t*/*q_t_* vs. *t* (do not differ significantly), it is indicative that the process mechanism follows the pseudo-second order model. The plot of *q_t_* versus *t*^0.5^ (according to Equation (5); is linear but does not pass through the coordinate origin, indicating that the intra-particle diffusion is not the mechanism of sorption. 

### 2.5. Properties of Loaded Adsorbent

#### 2.5.1. FTIR Analysis 

The FTIR spectra of adsorbent, before [26] and after adsorption were recorded. The FTIR spectra (Figure 6), showed the C-H stretching vibration of the methylene group is observed at 2834 and 2923 cm^−1^ in both the LMS-Fe_3_O_4_ and LMS-Fe_3_O_4_/Se samples [27]. The characteristic peaks of 2486 and 2359 cm^−1^ originated from O=C=O carbon-dioxide and amino N-H component, respectively [28,29]. The peak observed at 3292 cm^−1^, is ascribed to the N-H stretching vibration (primary and secondary amines) of LMS-Fe_3_O_4_. The band at 1475 cm^−1^ arises from the contribution of C-N bond stretching vibration [30]. The band 1640 cm^−1^ relates to the O-H bending vibration from Fe–OH; while the peak at 1376 cm^−1^ is assigned to the vibrations of lignin (macromolecule) units/rings, and C-O stretching vibrations [31]. The peak from C-N group vibration at 1037 cm^−1^ confirms a successful copolymerization of the LMS-Fe_3_O_4_. The lower wavelength bands, around 570 cm^−1^ (the peak of 566 cm^−1^), are ascribed to the stretching vibration mode associated to the metal-oxygen, Fe-O, in the crystalline lattice of LMS-Fe_3_O_4_ structure, respectively [32]. There were some band shifts after Se(VI) adsorption due to the formation of the new bonds such as the absorption bands at 1475 and 1376 cm^−1^ before adsorption was shifted to 1451 and 1350 cm^−1^ after adsorption. Similarly, the major FTIR absorption bands at 1640 and 1037 cm^−1^ remained unchanged before and after adsorbing 10 ppm of Se(VI), confirming the adsorbent stability.

#### 2.5.2. XRD Analysis 

Figure 7 shows the structural pattern of the A-LMS_Fe_3_O_4_ and A-LMS_Fe_3_O_4_/Se adsorbent, analyzed by the X-ray diffraction method. Diffractogram analysis indicates that there were slight changes between the initial and exhausted adsorbent. The XRD spectra display that the studied samples are heterogeneous, mainly containing lignin (broad peak observed at 2θ values of around 22 degrees, corresponding to the literature data [33]. The A-LMS_Fe_3_O_4_ sample contains magnetite (Fe_3_O_4_) in accordance with the ICDD card number 88-0315, where the dominant peak was observed at 2θ = 37°. 

The XRD spectrum of the A-LMS_Fe_3_O_4_/Se manifests a slightly weaker signal of the dominant peak, as well as of peaks for 2θ values of around 37 and 67 degrees, most likely as the consequence of the Se(VI) adsorption. The peaks at 2θ at the positions of 28.43, 32.68, 52.99, and 57.2°, attributed to the crystalline nature of sodium selenite [34], were not well-defined due to their overlapping with magnetite peaks.

#### 2.5.3. SEM/EDS Analysis

The SEM/EDS representative images of the A-LMS Fe_3_O_4_/Se adsorbent are shown in Figure 8. The adsorbent particles were spherical in shape, with a mean diameter of about 500 nm, high porosity, and coral structure. High porosity of the sample indicates the optimal copolymerization conditions and emulsifier content, which enable a high level of polymerization. The pore diameter was approximately 1 to 3 μm.

The SEM/EDS method determined the elemental composition of an adsorbent sample before and after adsorption of Se(VI) and the results, depending on the selected measuring point, are shown in Table 2 and Figure 9. The EDS spectra clearly indicate the domination of C and O, followed by Se, Fe, and Cl. The most intense peaks belong to the elements C and Cl. The presence of the Fe peak clearly indicates the incorporation of magnetite, and presence of the Se peak indicates the adsorption of Se(VI). 

### 2.6. Desorption Test—Regeneration Capacity 

Concentration and acidity/basicity of the regeneration medium are the main parameters which could contribute to the effective dislocation of the bonded oxyanion. The high amount of Se desorbed in 0.5 M NaOH indicates a regeneration capacity (E_des_) of 60.67%, using Equations (6) and (7). The concentration of desorbed Fe was 0.014 mg L^−1^ and Se was 1.98 mg L^−1^.

### 2.7. Real Water Samples

The Se removal efficiency by the A-LMS Fe_3_O_4_ adsorbent in real water samples is presented in Table 3. Several water samples were spiked so that the initial Se concentration in all samples was 10 mg L^−1^. The pH was kept to their natural origin (Table 3), to investigate more pragmatic practical usage—without a need for any chemical pretreatment or previous preparation. The Se removal efficiency was determined to be in the range of about 10–30%.

Respecting the principles of green chemistry, the application of concentrated chemical agents is not appropriate. On the other hand, the use of lignin-based adsorbent doped with the naturally occurred Fe-oxide, as applied in this investigation, is environmentally favorable. Connecting these two facts and taking into account a low sorbent regeneration ability, it is recommended to apply the spent sorbent as a source of Se in soil. Moreover, it is well known that in some countries, due to a low Se concentration in soil, it is necessary to add some Se, as fertilizer [35]. Therefore, the addition of spent lignin-based adsorbent into land and/or soil could be proposed. In this way, the single use of spent material in real conditions may be beneficial, to add Se in soil and to prevent spreading of secondary pollution. 

Thus, further research will be focused on the behavior of the spent sorbent in soil/land, primarily concerning Se mobility in real systems and the possibility of other pollutants (e.g., heavy metals) binding, to reduce their mobility in soil/land.

## 3. Discussion

Separation of selenium species onto various inorganic- and organic-based materials has been proven as an efficient water treatment (Table 4). The A-LMS Fe_3_O_4_ nanocomposite had higher capacities than the commercial magnetite. The reported uptake capacities for Se(VI) using the A-LMS Fe_3_O_4_ nanocomposite were according to the adsorption isotherm; the PFO and PSO values were 69.9 mg g^−1^, 29.64 mg g^−1^, and 41.56 mg g^−1^, respectively.

The other presented structures included varieties of commercial Fe_3_O_4_ [36], magnetic iron oxide nanoparticle/multi-walled carbon nanotubes [37], metal organic framework (MOF) [38,39], polymeric adsorbents [40], modified biochar [40,41,42], exfoliated kaolinite sheets/cellulose fibers nanocomposites [43].

**Table 4 ijms-23-13872-t004:** Comparison of different type nanocomposite materials and magnetite for adsorption Se(VI).

Adsorbent Type	Preparation/ Modification Method	pH	T (°C)	Concentration Se (mg L^−1^)	Adsorption Models	Kinetic Models	Adsorption Capacity (mg g^−1^)	Ref.
Fe_3_O_4_	Commercial	7	25	5	-	PFO	2.19	[36]
		7	25	5	-	PSO	0.21	
MIO-MWCNTs: magnetic iron oxide nanoparticle/multi-walled carbon nanotubes		1.8–7.1	15	5–100	-	PFO	3.799	[37]
			30				3.757	
			45				3.640	
		1.8–7.1	15	5–100	-	PSO	3.928	
			30				3.843	
			45				3.779	
Metal organic framework (MOF)								
Mercapto functionalized Zr-based magnetic metal–organic framework MUS: Fe_3_O_4_@SiO_2_@UiO-66-(SH)_2_(MUS)	Coprecipitation andsol–gel method	2	-	10–360	Langmuir	PSO	27.3	[38]
Binary MOFs, UiO-66(Fe/Zr)	Hydrothermal method	5	-	10–50	Langmuir	PSO	258	[39]
Poly(allyl trimethylammonium) grafted chitosan and biochar-BC composite (PATMAC-CTS-BC)	Polymerization process	5	25	10–50	Langmuir		98.99	[40]
						PFO	36.97	
						PSO	37.39	
Exfoliated kaolinite sheets/cellulose fibers nanocomposite (EXK/CF)	-	2	-	50	Langmuir		137.5	[43]
						PFO	74.5	
						PSO	88.5	
Iron-impregnated food waste biochar (Fe-FWB)	-	7	25	100–300	Langmuir		11.7	[42]
Magnetic biochar with magnetite nanoparticles MBC-SPS-450	-	5	23 ± 1	183	Freundlich		−98.03	[41]
		7					384.62	
		9					333.33	
Lignin microspheres modified with magnetite nanoparticles: A-LMS Fe_3_O_4_	Coprecipitation and copolymerization process	6.45	22	7.75	Adsorption isotherm		69.9	

						PFO	29.64	This study
						PSO	41.56	

The A-LMS Fe_3_O_4_ nanocomposite had very good capacities for the examined selenate relative to the reported materials in Table 4. This finding has reflected the suitability of composite as an advanced hybrid material with enhanced uptake capacities for the inorganic selenium as water pollutants. 

The surface sites activated by the Fe-oxide(s) were performed via different modification methods: hydrothermal method, coprecipitation, copolymerization and sol–gel method. Additionally, a magnetite-modified adsorbent may be efficiently separated with addition of an external magnetic field. 

Poly(allyl trimethylammonium) grafted chitosan and biochar composite were tested for SeO_4_^2−^ removal at a wide pH range from 2 to 10 [40]. Due to the permanent positive charges of quaternary ammonium groups (=N^+^-), the removal mechanisms of SeO_4_^2−^ were mainly attributed to the electrostatic interactions with =N^+^- and protonated -NH_3_^+^ groups, and redox-complexation interactions with -NH_2_, -NH-, and -OH groups. 

Guo and co-authors have reported that large specific surface area (467.52 m^2^ g^−1^) and uniform mesoporous structures of the synthesized metal organic framework (MOF) resulted in fast adsorption efficiency and high adsorption capacity for selenium species [39]. 

Mercapto-functionalized magnetic metal–organic framework: MUS (containing Fe_3_O_4,_ SiO_2,_ UiO-66-(SH)_2_) has shown improved removal of SeO_4_^2−^ because of a higher specific surface area with the abundant adsorption sites. Compared to the Cu/Co/Mn-Fe_2_O_4_ and Fe_3_O_4_/multi-walled carbon nanotube, the MUS has manifested a higher adsorption capacity for removal the inorganic Se species. The adsorption capacities of MUS were also proven for simultaneous removal of disparate species (mainly Se and Sb anions), with a high potential for practical applications [38].

Akukhadra and co-authors reported that the exfoliated kaolinite sheets/cellulose fibers nanocomposite (EXK/CF) was synthesized as a novel hybrid material of enhanced surface area and adsorption capacities for selenium. The combination of exfoliated kaolinite sheets and cellulose fibers as the natural biopolymers of improved physicochemical properties has resulted in a high adsorption capacity of Se ions for both organic and inorganic forms [43].

An important factor for adsorption techniques is based on the efficacy and effectiveness of the adsorbent—surface area, presence of functional groups onto the surface and strong affinity to adsorbate. Magnetite has a strong affinity for selenate, which is very important for the efficiency and effectiveness of the adsorbent.

## 4. Materials and Methods 

### 4.1. Material Synthesis and Design 

The highly porous A-LMS Fe_3_O_4_ adsorbent was synthesized and designed following a previously developed procedure [26,44].

Briefly, an amino-functionalized adsorbent (A-LMS) was obtained by the inverse suspension copolymerization of kraft lignin with the branched polyethylene-imine (PEI) [30], with the addition of epichlorohydrin as a cross-linker agent [24]. The preparation was performed as follows: 0.5 g of lignin and 4.0 g of NP-Fe_3_O_4_ was added to a three-necked flask (10 mL of demineralized water, DMW), then 2.0 g of PEI, 0.1 g of sodium dodecyl benzene sulfonate and 10 mL of sodium alginate emulsifier (5.0 wt%) solution were added with continuous stirring. The stimulation was continued at 22 °C (for 30 min), after which the temperature was raised to 60 °C. Liquid paraffin (oil phase) in volume of 80 mL was added to a flask to form a suspension, then 2.0 mL of the cross-linker, epichlorohydrin, was added drop-wise with continuous stirring for 120 min to complete the copolymerization process. After centrifugation, the obtained copolymerized A-LMS microspheres were washed with ether, ethanol, and water. The synthesized amino-modified lignin microspheres, A-LMS, were dried by the lyophilization for 24 h at −40 °C and further characterized [44].

### 4.2. Analytical Reagents and Determination of Concentration 

All solvents used in this study were of analytical grade. The Na_2_SeO_4_ (p.a. purity grade, Sigma–Aldrich, St. Louis, MO 63103, USA) was diluted with the DMW water (18 MΩ resistivity) in order to prepare the appropriate concentration for adsorption experiments. The NaOH solution (p.a. purity, Termohemija, Belgrade, Serbia) concentration of 0.5 mol L^−1^ was used for the desorption experiments. The pH of solutions was measured using a pH meter WTW 7310, Ino Lab. The Se concentration in solution was measured using an ICP-MS (model Agilent 7700, Agilent Technologies, Inc., Tokyo 192-8510 Japan) by the external calibration method. Experiments were performed in triplicate and the average values of a response function were taken into account in the statistical analysis. Interpretation and analysis of the results were also done using the Minitab software. Experiments were done in a random order to ensure that the results are not affected by the uncontrolled factors and experimental errors are properly estimated. The error bars are smaller than the symbols used in the figures. 

To check the experimental results, the statistical method of analysis of variance (ANOVA) was used in the software package OriginPro 8, in order to determine the kinetic parameters for the Se(VI) adsorption onto A-LMS Fe_3_O_4_ and based on this, the adsorption model.

### 4.3. Adsorption Experiment and Kinetic Study 

The adsorption of SeO_4_^2−^ anions was conducted in a batch system, with the initial concentrations of 7.75 mg L^−1^. The mass of the A-LMS-Fe_3_O_4_ adsorbent (0.025 mg) was added into 1 mL of the DMW and left overnight for 18 h (to stabilize material swelling). The suspension was added to 10 mL of the DMW (total volume of 11 mL), where the solution pH (6.48) and temperature (22 °C) were kept constant, while the agitation time (5, 10, 30, 60, 90, 120, 180, 240, 280, 300 min) varied. The reaction stimulation (shaking) was performed using an orbital Heldoph shaker (170 rpm). After the adsorption process, the samples were filtered through a 0.45-µm pore diameter membrane filter and acidified with nitric acid (1:1, *v/v*) before studying the analytical determination of ions.

The amount of Se adsorbed (mg g^−1^) at time *t* (*q_t_*) and in the equilibrium (*q*_e_) were calculated using the Equations (1) and (2), where *C*_0_, *C_t_* and *C*_e_ are the initial selenium concentration, selenium concentration in the solution after appropriate adsorption time (*t*) and in equilibrium (*C*_e_), respectively; *V* is solution volume and *m* is adsorbent mass.
(1)$$q_{t}=\left(C_{o}-C_{t}\right)\cdot V/m$$
(2)$$q_{e}=\left(C_{o}-C_{e}\right)\cdot V/m$$

The effect of contact time on the process efficiency (kinetic study) and adsorption capacity was carried out following the experimental conditions: *C_o_*(Se) = 7.75 mg L^−1^, pH of solution 6.48, temperature 22 °C and stirring rate 170 rpm (o min^−1^).

The relation between *q_t_* and time (*t*) was analyzed using the well-known kinetic models in linear forms:

Pseudo-first order [45,46]
(3)$$ln\frac{\left(q_{e}-q_{t}\right)}{q_{e}}=-k_{1}\cdot t\;\;$$

Pseudo-second order [46,47]
(4)$$\;\frac{t}{q_{t}}=\frac{1}{k_{2\cdot }q_{e}^{2\;}}+\frac{t}{q_{e}}\;$$

Intraparticle diffusion model [48,49]
(5)$$q_{t}=k_{d}\cdot t^{0.5}+c$$
where *k*_1_, *k*_2_, and *k_d_* denote constants of pseudo-first, pseudo-second and intraparticle diffusion model, respectively. The models’ parameters were calculated from the slope-intercept form. The correlation coefficient (*R*^2^) was calculated to evaluate the accuracy of applied models.

### 4.4. Material Characterization 

After the adsorption experiments, the structural, surface, and morphological properties of the saturated adsorbents were determined. The XRD analysis was conducted to identify the materials based on their diffraction pattern, and phase identification, thus assessing the crystallinity of the nanoparticles [50]. The XRD analysis was performed using a small-angle X-ray scattering (SAXS) diffractometer (Rigaku Smartlab, Austin, TX, USA) within the 2θ range 10–90 with 0.05 step size. 

The FTIR analysis was applied to confirm the presence of functional groups due to the applied surface-modification strategy. Surface functional groups of samples were determined using the Fourier-transform infrared spectroscopy—FTIR, within the range of 400–4000 cm^−1^, at 4 cm^−1^ of spectral resolution (type of instrument: BOMEM spectrometer, Bomem Inc./Hartman&Braun, Quebec, QC, Canada). Samples were made as the KBr pellets at 22 °C, and measurements were conducted in the range of 500–4000 cm^−1^. 

The surface morphology of the A-LMS Fe_3_O_4_ adsorbent was analyzed by the SEM/EDS method. The morphology of the adsorbent A-LMS Fe_3_O_4_ was studied using a scanning electron microscope (SEM model: JOEL JSM-IT300LV operated at 20 keV). The chemical composition of the samples was determined using an energy-dispersive X-ray spectroscopy (EDS). The EDS spectra were recorded using an X-ray spectrometer (Oxford Instruments, Abingdon, Wielka Brytania) attached to the scanning electron microscope and Aztec software. 

### 4.5. Desorption Test 

To investigate the affinity of the adsorbent regeneration, desorption tests were performed following the procedure: the exhausted A-LMS-Fe_3_O_4_ adsorbent was added to 1mL of DMW, and after 24 h 25 mL of concentrated NaOH (0.5 M) was added to the suspension. The contact time was 300 min, stirring 170 rpm. After the defined contact time, the concentration of desorbed Se was determined by the ICP-MS. 

The desorbed amount (*Q_des_*) is calculated as the amount of selenium desorbed from one gram of spent adsorbent (Equation (6)):(6)$$Q_{des}=c_{des}\cdot V_{des}/M$$
where *c_des_* is Se concentration in desorption solution, *V_des_* is the volume of desorption solution and *M* is the weight of spent adsorbent.

Finally, ghe desorption efficiency (%) is calculated as the ratio of desorbed amount (*Q_des_*) and initially sorbed amount *q_e_*, multiplied by 100 (Equation (7)):(7)$$E_{des}=\frac{Q_{des}}{q_{e}}\cdot 100\;\left(\%\right)$$

### 4.6. Adsorbent Application in Real Water (Potable) Samples

The potential application of designed A-LMS Fe_3_O_4_ was tested for Se removal in the real water samples. Six (6) potable water samples, originating from different municipalities in the Republic of Serbia, were tested. The Se was added (spiked) to each sample, at concentration of 10 mg L^−1^ and the adsorption was performed according to the previously described experimental conditions (Section 2.7). The experimental conditions were: contact time 300 min, stirring 170 rpm, adsorbent dose 0.23 g L^−1^, solution volume 11 mL. 

## 5. Conclusions

Based on the results concerning adsorption/desorption processes, the following conclusions were made:The removal efficiency of Se(VI) anions from the synthetic water samples was ≈99%, while for the real water samples, it was around 20%, indicating a competitive influence/effect of other ions present in water.The pseudo-second model was the most appropriate for the kinetic data description; adsorption capacity of investigated adsorbent towards Se(VI) was found to be 34.94 mg/g.The regeneration capacity of 61% in 0.5 M NaOH, as a desorption solution, was obtained in the first cycle.Despite that the desorption efficiency was relatively low, precisely such Se loaded sorbent could be used as a soil fertilizer, considering that minimal pre/post chemical treatment of this material is necessary.

Considering the previously derived conclusions, further investigation of the highly porous nature-based A-LMS Fe_3_O_4_ adsorbent will be directed at the simultaneous removal of SeO_4_^2^, AsO_4_^2−^ and CrO_4_^2−^ anions and the effect of the co-existing ions on the overall removal process in real water samples. 

## Figures and Tables

**Figure 1 ijms-23-13872-f001:**
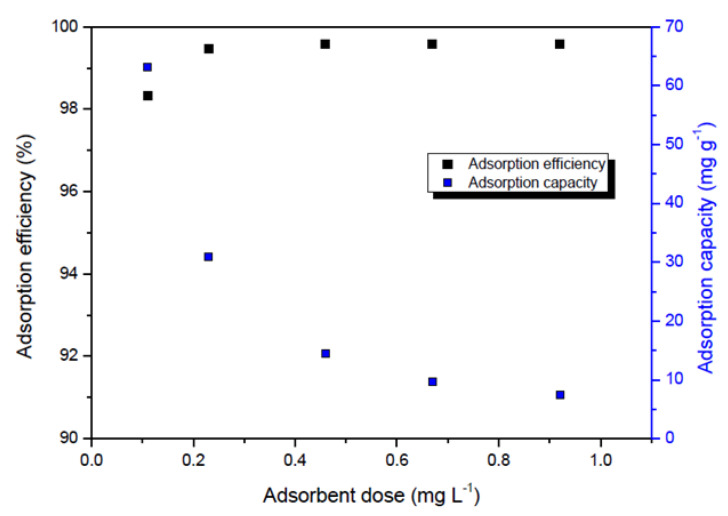
Adsorption efficiency/capacity versus the adsorbent dose.

**Figure 2 ijms-23-13872-f002:**
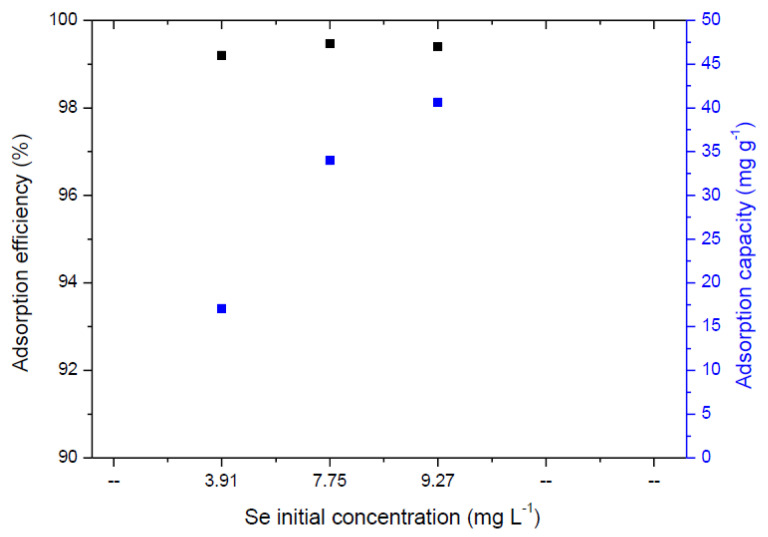
The effect of selenium initial concentration on adsorption efficiency/capacity.

**Figure 3 ijms-23-13872-f003:**
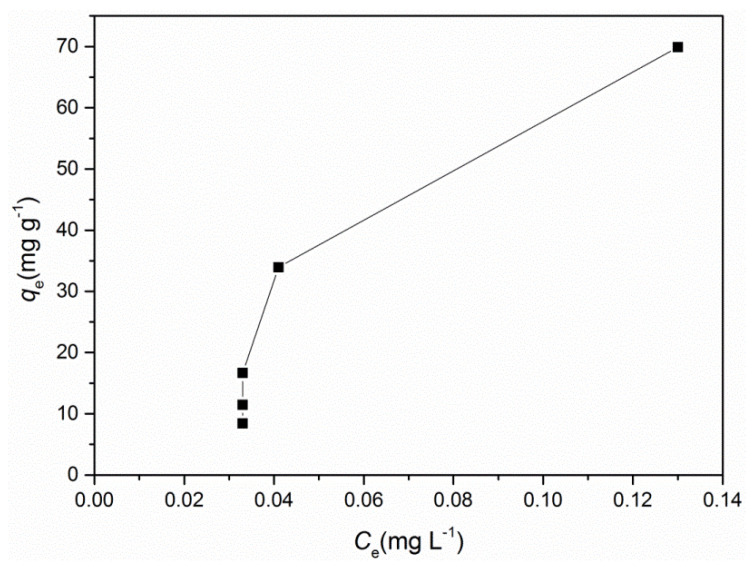
The adsorption isotherm for selenate by the A-LMS Fe_3_O_4_ adsorbent; experimental conditions: the adsorbent dose was set at 0.23 g L^−1^, pH 6.46, temperature 22 °C and stirring 170 rpm.

**Figure 4 ijms-23-13872-f004:**
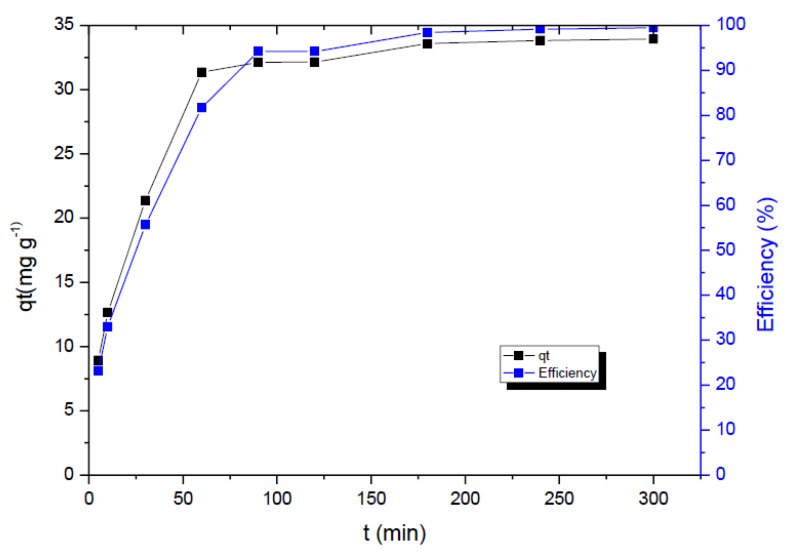
The effect of contact time onto adsorption efficiency (%) and capacity (mg g^−1^); experimental conditions: *C*_i_ = 7.75 mg L^−1^, adsorbent dose 0.23 g L^−1^, pH 6.46, temperature 22 °C, stirring 170 rpm.

**Figure 5 ijms-23-13872-f005:**
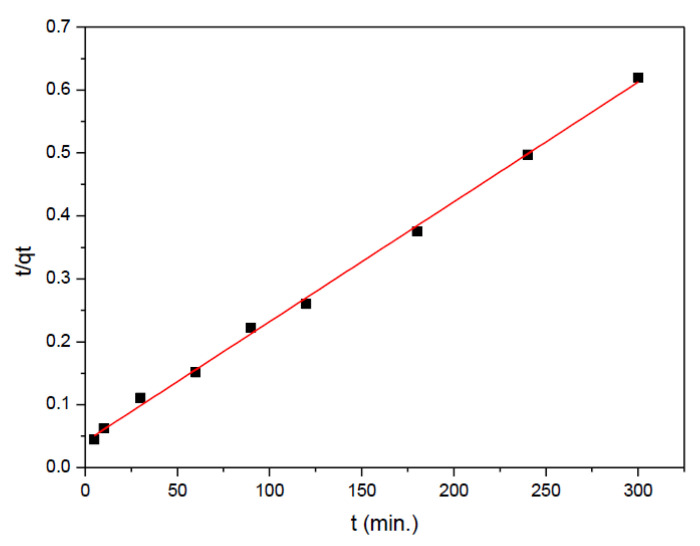
Pseudo second-order kinetic plots for adsorption of selenate onto A-LMS Fe_3_O_4._

**Figure 6 ijms-23-13872-f006:**
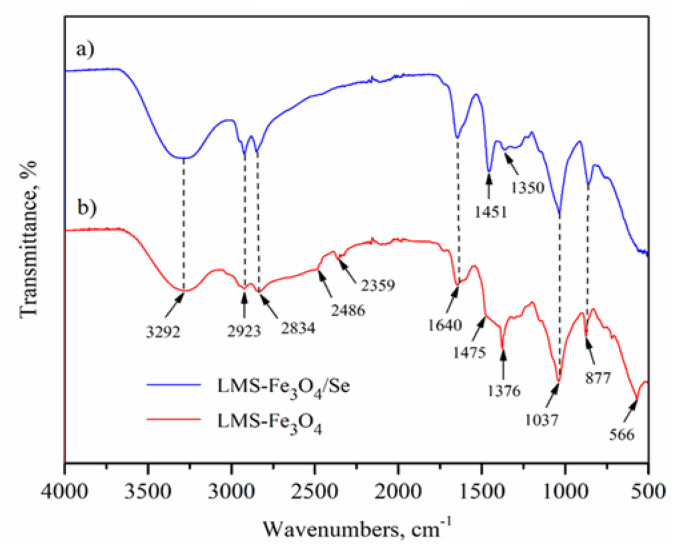
The FTIR spectrum of Se loaded A-LMS Fe_3_O_4_ adsorbent, (**a**) after adsorption; (**b**) before adsorption

**Figure 7 ijms-23-13872-f007:**
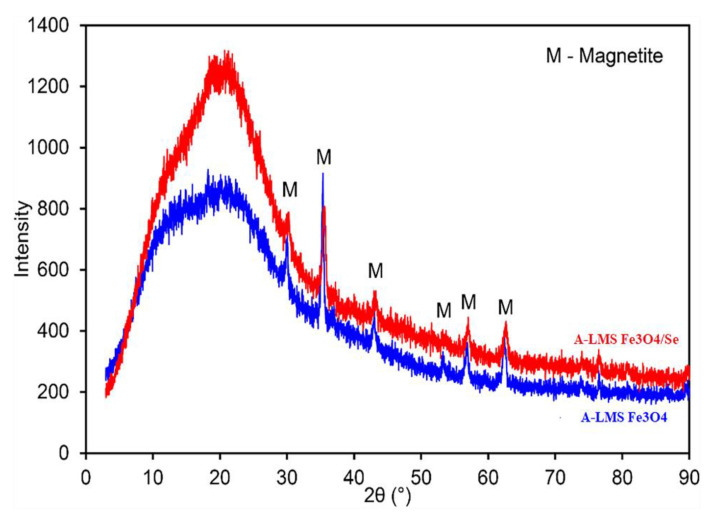
XRD analysis of the A-LMS Fe_3_O_4_ (blue) and A-LMS Fe_3_O_4_/Se (red) adsorbent.

**Figure 8 ijms-23-13872-f008:**
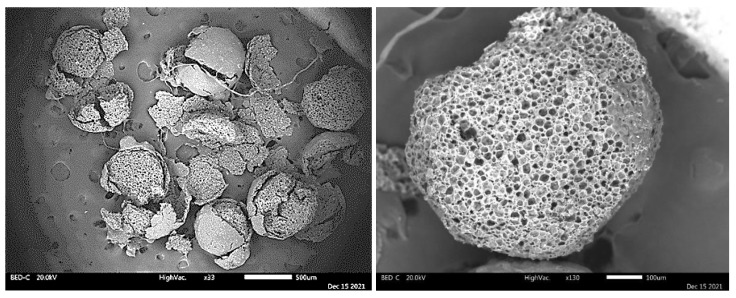
SEM images of the A-LMS Fe_3_O_4_/Se adsorbent.

**Figure 9 ijms-23-13872-f009:**
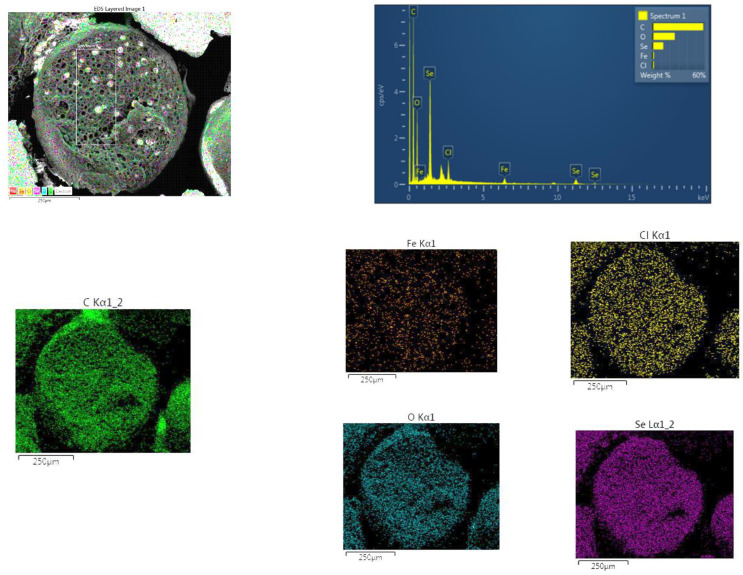
EDS image with map of element distribution of the A-LMS Fe_3_O_4_/Se adsorbent.

**Table 1 ijms-23-13872-t001:** Kinetic parameters for the Se(VI) adsorption onto A-LMS Fe_3_O_4._

Models	Pseudo-First Order Model (PFO)			Pseudo-Second Order Model (PSO)			Intraparticle Diffusion Model		
**Parameters**	*k*_1_ (min^−1^)	*q_e_* (mg g^−1^)	*R* ^2^	*k*_2_ (g mg^−1^ min^−1^)	*q_e_* (mg g^−1^)	*R* ^2^	*k_id_*_1_ (mg g^−1^ min^−1/2^)	*C*_1_ (mg g^−1^)	*R* ^2^
	0.022	29.64	0.976	0.014	41.56	0.998	1.976	9.820	0.949

**Table 2 ijms-23-13872-t002:** The content of elements (%) present on the A-LMS Fe_3_O_4_/Se surface.

Spectrum Label	C, %	O, %	Cl, %	Fe, %	Se, %	Total, %	Figure
Spectrum 1 (1/x160)	58.39	25.69	1.70	1.79	12.43	100.00	Figure 8
Spectrum 2 (1/x130)	60.87	24.11	1.72	0.74	12.28	100.00	Appendix A
Spectrum 3 (1/x130)	61.20	20.18	1.76	4.59	12.27	100.00	Appendix A
Spectrum 6 (1/x130)	61.62	19.21	2.72	0.70	15.75	100.00	Appendix A
Spectrum 7 (1/x130)	61.06	24.21	1.88	0.90	11.94	100.00	Appendix A

**Table 3 ijms-23-13872-t003:** The adsorption capacity of the A-LMS Fe_3_O_4_ for Se removal from spiked drinking water.

Sample	*q*_e_, mg g^−1^	pH_i_
1	14.08	7.67
2	8.80	7.32
3	7.48	7.48
4	6.16	7.42
5	4.94	7.21
6	4.84	7.00

## Data Availability

Not applicable.

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
