# Peer review of "Lignin Microspheres Modified with Magnetite Nanoparticles as a Selenate Highly Porous Adsorbent"

_ijms, 2022, doi:10.3390/ijms232213872_

Round 1

Reviewer 1 Report

In this manuscript by Marjanovic et al, the authors demonstrate the use of lignin-based microspheres modified with magnetite nanoparticles as an adsorbent of selenate from lab, as well as real water samples from Serbia spiked with Se. The adsorbent samples are synthesized by the authors and the capacity for Se removal is studied in terms of several factors including, the amounts of adsorbent and adsorbate, and kinetics of adsorption. The adsorbent samples before and after adsorption are characterized using FTIR. XRD and SEM-EDS. While the objective of the work is important and of relevance, the work itself and the manuscript suffers from several shortcomings making it unsuitable for publication in this journal in the current form. I therefore do not recommend accepting this work for publication in this journal unless the authors are willing to revisit the work and undertake a major revision of the manuscript. Listed below are some specific comments that will help the authors revise the manuscript.

1. It seems the authors took single measurements and based the inferences on them leaving no room for an uncertainty analysis. Ideally, it is helpful to take 3-5 measurements of the quantities reported here (e.g. adsorption amounts) and report the mean values with the standard deviation estimated as uncertainty/errors. This gives credence to the study. However, no uncertainty/error analysis is reported by the authors. The authors should consider including such an analysis by taking more measurements.

2. Just as no error analysis is reported as mentioned above, the authors have not put the study in perspective by comparing there results with the literature. For example, the authors report the maximum adsorption capacity as 69.9 mg/g. How good or poor is this capacity compared to other adsorbents (e.g. magnetite nanoparticles)? In other words, does adding magnetite nanoparticles to lignin improve the adsorption capacity of the former? In absence of such a discussion, the study in itself is not so useful.

3. How do the authors define adsorption efficiency shown in Figures 1 and 2? It would improve the readability of this quantity in these figures if the corresponding Y range is shortened to perhaps 90 - 100 % instead of 0 - 100 % shown in the present manuscript.

4. The authors report the kinetics of adsorption in Figure 4. The number of data points is really very small (just 4). The authors should consider measuring the adsorption capacity at more number of contact times.

5. In the FTIR spectra shown in Figure 6, peaks at 2486, 2359 and 566 cm^-1 in the adsorbent before adsorption are missing in the spectra after adsorption. What do these peaks represent?

6. SEM images shown in Figure 8 do not seem to match the description that appear between lines 175 and 179. For example, line 176-177, the authors say that the adsorbent particles are spherical with size about 500 nm. However, with the scale in the figures showing 100 (right panel) or 200 (left panel) microns, it is not possible to see anything that measures 500 nm (0.5 microns).

7. Finally, the title of the manuscript seems to emphasize 'high porosity' of the adsorbent. While the adsorbent might indeed be highly porous, this characteristic seems to have had no effect on the study here, or at least the authors have not discussed it in enough details.

Reviewer 2 Report

In the article entitled "Lignin Microspheres Modified with Magnetide Nanoparticles as Selenate Highly Porous Adsorbent" written by Marjanovic et al. the examination of the amino-functionalized lignin microspheres (A-LMS) modified with magnetite nanoparticles (NP-Fe3O4) was performed with the use of FTIR, XRD and SEM techniques. In general, the manuscript is well written and structured. The methodology is adequate and explicitly stated. The quality and quantity of presented data are completed.I have few remarks as follows:

1.) Line 62 - Saccharomyces cerevisiae, Ganoderma lucidum, Ascomycota should be written in italics

2.) In Introduction section some information about analytical techniques - FTIR, XRD and SEM should be included

3.) Results and Discussion should be devided into two separate sections.

4.) The Discussion seems good but need to go in depth, please revise.

5.) Conclusion: This section needs to be more concise and shortened.

6.) I recommend reviewing the bibliography and making a selection of the references to remove the older ones (from last five years, period 2016-2022).

Round 2

Reviewer 1 Report

In this revision, the authors undertook a major revision of their work and have addressed most of the comments made by the reviewers in the previous round of review. The resulting manuscript is now suitable for publication. I just have one minor comment that the authors may address before publication.

In the methods section, the authors state that they took measurements in triplicate and carried out a proper estimation of uncertainties involved in the measurements. However, no error estimates are reported in the results. For example, figures still show only symbols and no error bars. Are the error bars smaller than the symbols used? If yes, the authors should include a sentence stating that in the captions. Or, if the authors have simply not included the error bars in the figures, they should be included now.
